# Social Learning of Acquiring Novel Feeding Habit in Mandarin Fish (*Siniperca chuatsi*)

**DOI:** 10.3390/ijms20184399

**Published:** 2019-09-07

**Authors:** Jian Peng, Ya-Qi Dou, Hui Liang, Shan He, Xu-Fang Liang, Lin-Jie Shi

**Affiliations:** 1College of Fisheries, Chinese Perch Research Center, Huazhong Agricultural University, Wuhan 430070, China; 2Freshwater Aquaculture Collaborative Innovation Center of Hubei Province, Wuhan 430070, China; 3Key Lab of Freshwater Animal Breeding, Ministry of Agriculture, Wuhan 430070, China

**Keywords:** social learning, feeding habit, domestication, mandarin fish, c-Fos, appetite control gene

## Abstract

Social learning plays important roles in gaining new foraging skills and food preferences. However, the potential role and molecular mechanism of social learning in acquiring new feeding habits is less clear in fish. In the present study, we examined the success rate of feeding habit domestication from live prey fish to dead prey fish, as well as the food intake of dead prey fish in mandarin fish with or without feeders of dead prey fish as demonstrators. Here, we found that mandarin fish can learn from each other how to solve novel foraging tasks, feeding on dead prey fish. In addition, the analysis of gene expressions and signaling pathways of learning through Western blotting and transcriptome sequencing shows that the expression of the *c-fos, fra2, zif268, c/ebpd* and *sytIV* genes were significantly increased, and the anorexigenic *pomc* and *leptin a* expressions were decreased in fish of the learning group. The phosphorylation levels of protein kinase A (PKA) and Ca^2+^/calmodulin-dependent protein kinase II (CaMKII) in the learning group were significantly higher than those of the control group, while the phosphorylation level of S6 ribosomal protein (S6) was lower. With the inhibitors of PKA and CaMKII signaling and the chromatin immunoprecipitation (ChIP) assay, we further found that the social learning of new feeding habits in mandarin fish could be attributed to the activation of the CaMKII signaling pathway and then the stimulation of the expression of the *c-fos* gene, which might be an important transcriptional factor to inhibit the expression of the anorexigenic gene *pomc*, resulting in the food intake of dead prey fish in mandarin fish. Altogether, our results support the hypothesis that social learning could facilitate the acquisition of novel feeding habits in fish, and it considerably increases the rate of subsequent individual food intake and domestication through the interaction between the learning gene *c-fos* and the appetite control gene *pomc*.

## 1. Introduction

Social learning is the process in which animals gain new skills and information through interaction with or observation of other individuals, and it eliminates the need for individuals to pay the costs of acquiring information on their own. Social learning plays important roles in a wide range of contexts, including acquiring new foraging skills and food preferences. There is now strong experimental evidence in humans that the eating behavior of other people can strongly influence the amount of food that we eat [1,2]. The learning of foraging what, where and how to eat is the main topic in studies of behavioral innovations in birds [3,4]. 

Because of the complex foraging environments and the change of food items, seasonally or spatially, learning plays a role in the foraging flexibility of fish [5]. Several studies have demonstrated that experience influences the foraging success, novel foraging behavior and profitability of different prey types. Groups of fish forage more efficiently than individuals, such as the goldfish (*Carassius auratus auratus*), bluntnose minnows (*Pimephales notatus*), Alaska pollock (*Theragra chalcogramma*) and sticklebacks (*Gasterosteus aculeatus*) [6,7,8,9]. The enhancement of foraging by social learning has also been reported in hatchery-reared juvenile chum salmon (*Oncorhynchus keta*), Alaska pollock and rock bass (*Ambloplites rupestris*) [8,10,11]. In addition, there are well-documented effects of social learning on novel foraging behavior through the observation of conspecifics in fishes. Anthouard reported that European sea bass (*Dicentrarchus labrax*) learned to press a lever to receive a food reward through the observation of the trained demonstrators [12]. However, little is known about the potential role of social learning in acquiring new feeding habits in fish.

Mandarin fish (*Siniperca chuatsi*) is an important aquaculture fish and highly valued due to its excellent flesh quality with an output value of more than 3 billion U.S. dollars (20 billion Yuan). Mandarin fish are aggressive piscivores; once their fry start feeding, they feed solely on live fry of other fish species and refuse zooplankton or formulated diets [13]. The acceptance of dead prey fish or artificial diets represents a key challenge in the aquaculture of mandarin fish, similar in the Eurasiam perch [14]. We designed a specific training paradigm to domesticate the feeding habits of mandarin fish, and we found that some individuals could accept dead prey fish successfully (feeders) [15]. What is less clear is whether mandarin fish are aware of the influence of social learning from feeders on their feeding behavior of accepting dead prey fish as a novel feeding habit.

Observational learning, i.e., the learning of individual (observer, actor) by the direct observation of the behavior of an experienced individual (demonstrator, tutor) is the most frequent type of social learning [16]. To investigate the effect of social learning on acquiring new feeding habits of mandarin fish, we examined the success rate of feeding habit domestication from live prey fish to dead prey fish, as well as the food intake of dead prey fish in mandarin fish with or without feeders of dead prey fish as demonstrator. In addition, the analysis of gene expressions and signaling pathways of learning through Western blotting, transcriptome sequencing, inhibitor treatment and a chromatin immunoprecipitation (ChIP) assay allowed us to gain insights into the molecular mechanism of social learning in acquiring new feeding habits in mandarin fish. These results can support aquaculture domestication programs and research in mandarin fish, and they can lay a solid genetic foundation to understand the social learning of feeding behavior.

## 2. Results

### 2.1. Success Rate, Learning Times, and Food Intake of Dead Prey Fish

The success rate of feeding habit domestication from live prey fish to dead prey fish in the learning group (with the positive demonstration fish) was significantly higher than in that of the control group (with the negative demonstration fish), and the learning times were significantly decreased in the learning group (Figure 1). The food intake of dead prey fish was also increased in the learning group (Figure 1). Mandarin fish with the positive demonstration fish might be easier to be domesticated to accept dead prey fish, suggesting the social learning of acquiring new feeding habits in mandarin fish. 

### 2.2. Expression of Learning or Appetite Control-Relative Genes

As shown in Figure 2, with the positive demonstration fish, the gene expressions of *c-fos*, *fos-related antigen 2* (*fra2*), *zif268*, *ccaat/enhancer binding protein d* (*c/ebpd*) and *synaptotagmin-4* (*sytIV*) were significantly increased (*p* < 0.05) in fish of the learning group, whereas the expression of the *cAMP-response element binding protein I* (*creb I*) and *brain-derived neurotrophic factor* (*bdnf*) genes showed no difference between the two groups. For the appetite control genes, we examined the orexigenic genes such as *neuropeptide y* (*npy*), *agouti-related protein* (*agrp*) and *ghrelin*, and the anorexigenic genes such as *pomc*, *cocaine and amphetamine-regulated transcript* (*cart*), *leptin a* and *cholecystokinin* (*cck*) in the two groups. We found that the orexigenic gene *agrp* and the anorexigenic genes *pomc* and *leptin a* were significantly decreased in the fish of learning group (*p* < 0.05), while no changes were found in the expression of the *npy*, *ghrelin*, *cart* and *cck* genes between the two groups (Figure 3). 

### 2.3. Transcriptome Analysis

To obtain an overview of the gene expression profile in mandarin fish with or without social learning, cDNA libraries were constructed from the brains of mandarin fish with the negative demonstration fish (wild mandarin fish without training) (Group CG) and fish with the positive demonstration fish (well-trained fish which could accept dead prey fish) (Group LG), which were sequenced using a BGISEQ (Beijing Genomics institution sequencing)-500 system. After removing the low-quality reads, we obtained 64,448,976 (CG) and 64,560,824 (LG) clean reads (Table 1). The removal of partial overlapping sequences yielded 93,699 distinct sequences (All-Unigene, mean size: 1049 bp, N50: 1775 bp). Among these unigenes, 69.5% (82,108) were between 100 and 500 bp in length, and 30.6% (36,110) were longer than 500 bp, of which 9.8% (11,550) were longer than 1000 bp. The sequencing data in this study have been deposited in the Sequence Read Archive (SRA) at the National Center for Biotechnology Information (NCBI) (accession number: PRJNA529539). Of the 67,206 annotated sequences in the mandarin fish transcriptome, 47.65%, 69.24%, 39.10%, 39.73%, 34.47%, 27.80% and 8.97% were assigned with the NR, NT, SwissProt, KEGG, KOG, Pfam and GO databases, respectively.

We found 24,819 genes to be differentially expressed between Groups CG and LG (FDR (false discovery rate) ≤ 0.001, fold-change ≥ 2). We mapped the differentially expressed genes to the reference canonical pathways in KEGG to identify the biological pathways. The representative pathways with the differentially expressed genes were cyclic AMP (cAMP) signaling, neurotrophin signaling, long-term potentiation, long-term depression, serotonergic synapse, dopaminergic synapse, glutamatergic synapse, and cholinergic synapse pathways (Figure 4, Figure 5, Figure 6, Figure 7, Figure 8, Figure 9, Figure 10 and Figure 11, Appendix A).

### 2.4. Learning Related Signaling Pathways 

For the signaling of the learning activated, we examined the phosphorylation level of AMP-response element-binding (CREB), protein kinase A (PKA), extracellular regulated protein kinases (ERK)1/2, protein kinase B (AKT), Ca^2+^/calmodulin-dependent protein kinase II (CaMKII) and S6 ribosomal protein (S6), which have been reported to be involved in the gene activation of learning or appetite control. The phosphorylation levels of PKA and CaMKII in fish of the learning group were significantly higher than those of the control group, while the phosphorylation level of S6 was lower than that of the control group, and the phosphorylation level of CREB, ERK1/2 and AKT showed no difference between the two groups (Figure 12).

### 2.5. PKA and CaMKII Signaling Pathway

To confirm whether the PKA and CaMKII signaling pathways are involved in feeding habit domestication through social learning, we used the specific inhibitors to study the effects of PKA and CaMKII signaling on the regulation of the expression of learning and appetite control genes. No significant change of the p-PKA level and mRNA expression of the *zif268*, *c/ebpd*, *npy*, *pomc*, *leptin a* and melanin-concentrating hormone (*mch*) genes were observed upon the inhibitor H-89 treatment, but the phospho-cAMP-response element binding (p-CREB) level, *c-fos* and *agrp* gene expressions were significantly decreased (*p* < 0.05) (Figure 13). As shown in Figure 14, we found that the p-CaMKII level and *c-fos* expression with KN-62 treatment were significantly lower than that without treatment (*p* < 0.05), the p-CREB level and *pomc* expression were significantly increased (*p* < 0.05), and the expression of the *agrp* and *leptin a* genes show no changes between groups.

### 2.6. ChIP

To further determine whether c-Fos has any direct regulation on the transcriptional activity of the *pomc* gene, we examined the interaction between transcriptional factor c-Fos and the three potential AP-1 (activator protein 1) binding sites of the *pomc* gene with a ChIP assay. We found the significant enrichments at Site 3 in c-Fos-immunoprecipitate compared with the IgG control (Figure 14), suggesting the binding of c-Fos to AP-1 binding sites in the regulatory region of the *pomc* gene.

## 3. Discussion

Most studies of social learning in a foraging context have been focused on the social transmission of new foraging skills [17,18,19] and food preferences [20,21,22]. To date, there have been few studies demonstrating learning processes in the feeding habit domestication of fish. Our study showed that mandarin fish were capable of social learning to acquire a new feeding habit. The success rates of feeding habit domestication from live prey fish to dead prey fish was significantly higher in mandarin fish with the positive demonstration fish than that with the negative demonstration fish. The food intake of dead prey fish was also increased in the learning group, and the learning times were decreased. Mandarin fish with the positive demonstration fish might be easier to be domesticated to feed on dead prey fish, suggesting that the learning ability of mandarin fish during the domestication of feeding habits. This is the first time that it has been shown that that social learning could enable mandarin fish with a naive feeding habit of live prey fish to acquire information of accepting dead prey fish from a knowledgeable one, thus avoiding the costs of learning solely by individual experience. Food preference resulting from the observation of the feeding behavior of the demonstrator has been found also in other fish. The observation of a trained conspecific Atlantic salmon *Salmo salar* significantly increased the rate at which naive hatchery-reared fish accepted novel, live prey items [23]. The young-of-the-year perch *Perca fluviatilis* could finish a shift from natural food items to commercial dry feeds faster through the presence of conspecifics that are familiar with the dry feed than without the demonstrators [24]. Previous studies have shown that changes due to experience increased the capture rate of a specific prey type and shortened the time to approach a novel food item in naive rainbow trout (*Oncorhynchus mykiss*) [25] and pink salmon (*Oncorhynchus gorbuscha*) [26]. Our results indicated that social learning could facilitate the acquisition of a novel feeding habit in mandarin fish and have a great effect on domestication and food intake. However, it is unclear how social learning could affect feeding habits, as is the molecular mechanism of social learning on food preference.

With regard to the molecular mechanism of social learning during the domestication of feeding habits in mandarin fish, we examined the mRNA expression of learning genes. In mandarin fish with the positive demonstration fish, the gene expressions of *c-fos*, *fra2*, *zif268*, *c/ebpd* and *sytIV* were significantly increased, and the expression of the *creb I* and *bdnf* genes showed no difference between the two groups. The TORC1 (target of rapamycin complex 1)-mediated CREB regulation is a critical molecular step underlying synaptic plasticity and long-term memory [27]. As members of immediate early gene (IEG) and the Fos family of transcription factors, *c-fos* and *fra2* mRNA expressions are up-regulated in response to a variety of neuronal activation protocols, including long-term protentiation (LTP) [28] and behavioral training [29,30]. The expression of the c-Fos protein is increased in the hippocampus of rats trained on the socially transmitted food preference (STFP) [31]. The short-term memory impairment of ethanol-exposed rats is related to a decrease of *c-fos* expression in the hippocampus [32]. Besides *c-fos*, another IEG gene, *zif268,* is reported to be involved in learning and memory, and several studies have shown that *zif268* mRNA is upregulated during different forms of associative learning [33,34]. In the present study, compared with fish in the control group, the mRNA levels of *c-fos*, *zif268* and *c/ebpd* were dramatically increased in fish of the learning group, suggesting that the social learning to exploit new food sources in mandarin fish might be attributed to the higher expression of the *c-fos*, *zif268* and *c/ebpd* genes, potentially resulting in the long-term protentiation of a new feeding habit (feed on dead prey fish).

Consistent with the increase of food intake of dead prey fish, mandarin fish with the positive demonstration fish showed a decrease of anorexigenic *pomc* and *leptin a*. The neuropeptides *npy*, *agrp*, *pomc* and *cart* play essential roles in the regulation of food intake and energy homeostasis in mammals [35]. Recent findings have demonstrated that peripherally-derived endocrine signals act on receptors in hippocampal neurons to regulate food intake through leptin or ghrelin and to learn food reward-driven responding [36]. Our results suggested that the lower expression of the anorexigenic genes *pomc* and *leptin a* might facilitate the food intake of dead prey fish in the learning group. Furthermore, c-Fos has been considered as the learning molecule, whether the appetite control genes *pomc* and *leptin a* could be the downstream molecular targets of c-Fos is poorly understood.

Though social learning is necessary for the domestication of feeding habits, the neural mechanisms are mostly unknown. To obtain an overview of gene expression profile in mandarin fish with or without learning, we found the differentially expressed genes between Groups CG and LG in several signaling pathways, such as the cAMP signaling, neurotrophin signaling, long-term potentiation, long-term depression, serotonergic synapse, dopaminergic synapse, glutamatergic synapse, and cholinergic synapse pathways. Learning and memory processes depend on electro-chemical signaling undertaken by amino acids, biogenic monoamines, acetylcholine, gasotransmitters, and peptides within the neuronal networks in the brain [37]. It has been shown that dopamine is involved in the expression of activity-dependent synaptic plasticity as well as in behavioral learning and learning-associated immediate-early gene expression [38]. In the present study, mandarin fish in the LG group showed a higher expression of D3-class receptors (D3/D4R), suggesting that the exposure to the novel feeding habit of dead prey fish might be known to release dopamine and facilitate dopamine-dependent synaptic plasticity induction via D3-class receptors. The efficacy of glutamate transmission can be persistently changed following the neuronal activities [39] in LTP and long-term depression (LTD) [40]. Our results also showed that the glutamatergic synapse signaling was involved in the social learning process of new feeding habits in mandarin fish through both long-term potentiation and long-term depression. 

Here, to investigate the signaling pathways that lead to activation of the IEGs and the upstream targets of these genes will advance our understanding of how functional activation of *c-fos* and *zif268* may be implicated in processing social learning. It remains unclear why the social learning of feeding habits in mandarin fish promoted the expression of learning related genes, such as *c-fos*, *fra2*, *zif268*, *c/ebpd* and *sytIV*. The long-lasting synaptic plasticity, which is frequently investigated in LTP and in learning paradigms, is associated with several cellular events: Ca^2+^ influx through the N-methyl-D-aspartate (NMDA) receptor, the generation of cAMP and the activation of PKA, the phosphorylation of mitogen-associated protein kinase (MAPK) and CREB, and the subsequent transcription of plasticity-associated genes [28,41]. CaMKII is also an essential component of the signaling cascade required for the regulation of synaptic plasticity and neuronal growth [42]. The PKA signaling pathway has been proposed to participate in both the late phase of long term potentiation and protein synthesis-dependent phase of memory formation [43]. Based on the current study, the increased success rate of domestication and food intake of dead prey fish could be attributed to the phosphorylation and activation of the PKA and CaMKII signaling pathway. The learning-induced phosphorylation of CREB co-occurs with the expression of the c-Fos protein [31]. There strong evidence that the phosphorylation of CREB is important for the long term consolidation of memories and long-term memory formation [44,45], but not short-term memory for socially transmitted food preference [31]. In the present study, p-CREB was measured in the brain of mandarin fish following the acquisition of a new feeding habit. The result indicated that the phosphorylation of CREB is not related to the short-term memory of new feeding habits in mandarin fish. In addition, the mechanistic target of rapamycin (mTOR) signaling is considered as an intracellular signaling for the regulation of food intake and energy homeostasis [46]. We found that the phosphorylation level of the S6 ribosomal protein (p-S6) was significantly decreased in the learning group with the increased food intake of dead prey fish. The result was consistent the hypothesis that the inhibition of mTOR signaling might stimulate food intake through changing the translation levels of the anorexigenic Pomc protein in mandarin fish (data not published). 

The interaction between learning genes and appetite control genes involved in the social learning of mandarin fish to accept dead prey fish through the PKA and CaMKII signaling pathways remains to be further explored. We used the inhibitors of the learning signaling pathway, H-89 for PKA and KN-62 for CaMKII, to confirm if the learning signaling pathway, especially the learning gene *c-fos*, could affect appetite control genes. The results showed that no significant change of the p-PKA level and mRNA expression of the *zif268*, *c/ebpd*, *npy*, *pomc*, *leptin a* and *mch* genes were observed upon the inhibitor H-89 treatment, but the p-CREB level, *c-fos* and *agrp* gene expressions were significantly decreased. Though the p-PKA level did not change with inhibitor treatment, the downstream signaling of the PKA pathway, the p-CREB level was decreased, suggesting that the PKA pathway might regulate *c-fos* expression through CREB signaling but not work on the anorexigenic gene expression. The p-CaMKII level and *c-fos* expression with KN-62 treatment were significantly lower than that without treatment, the p-CREB level and *pomc* expression were significantly increased, and the expression of the *agrp* and *leptin* a genes show no changes between groups. It is suggested that the CaMKII signaling pathway activated by social learning could stimulate the expression of the *c-fos* gene, and then c-fos might be an important transcriptional factor to inhibit the expression of the anorexigenic gene *pomc*, resulting the increase of food intake of dead prey fish in mandarin fish.

To determine whether c-Fos has any direct regulation on transcriptional activity of the *pomc* gene, we examined the interaction between transcriptional factor c-Fos and the three potential AP-1 binding sites of the *pomc* gene with a ChIP assay. We found significant enrichments at the Site 3 in c-Fos-immunoprecipitate, suggesting the binding of c-Fos to AP-1 binding sites in the regulatory region of *pomc*. A previous study has shown that c-Fos functions as a proapoptotic agent by binding the c-FLIP(L) (FLICE (FADD-like IL-1β-converting enzyme) -like inhibitory protein) promoter and repressing the antiapoptotic molecule c-FLIP(L) transcriptional activity [47]. The AP-1 family member protein c-Fos plays a crucial role in a variety of biological processes, and the downstream targets of c-Fos have been identified in a wide range of normal development, inflammation, and oncogenesis [47]. In this report, we showed that c-Fos, in addition to its well-known oncogenic function, has a novel regulatory function on food intake by the *pomc* gene. The regulation of c-Fos on *pomc* might be a key pathway for feeding and food habit modification by learning, and the activator of the c-Fos/*pomc* pathway could play important roles in animal culture with artificial diets, especially endangered species and economic animals, and in the formation of healthy food preferences for children, such as socially transmitted food preference.

In conclusion, we investigated the effect of social learning on the feeding habit domestication from live prey fish to dead prey fish in mandarin fish. Our results showed that with the positive demonstration fish that feed on the dead prey fish, mandarin fish also could accept the dead prey fish with a higher success rate and lower learning times than those with the negative demonstration fish. The social learning of acquiring new feeding habits in mandarin fish could be attributed to the expression of learning and appetite control genes, as well as the activation of the CaMKII signaling pathway. The CaMKII signaling pathway was activated by social learning and the stimulation of the expression of the *c-fos* gene. As such, the *c-fos* gene might be an important transcriptional factor to inhibit the expression of the anorexigenic gene *pomc*, resulting in an increase of the food intake of dead prey fish in mandarin fish. This offers a first insight into the ability and molecular mechanism of social learning on acquiring novel feeding habits in fish. As the ability to copy the feeding behavior from intraspecific tutors, our experiment encourages further studies on the potential roles of social learning in the context of feeding training in fish, especially for large-scale aquaculture purpose.

## 4. Materials and Methods

### 4.1. Reagents and Fish

Phospho-CREB (Ser133) (87G3) Rabbit mAb, Phospho-PKA C (Thr197) (D45D3) Rabbit mAb, Phospho-p44/42 MAPK (Erk1/2) (Thr202/Tyr204) (D13.14.4E) XP® Rabbit mAb, Phospho-CaMKII (Thr286) (D21E4) Rabbit mAb, Phospho-S6(Ser235/236) (D57.2.2E) XP® Rabbit mAb and H-89 (Dihydrochloride) were purchased from Cell Signaling Technology (Beverly, MA, USA). Phospho-AKT (Thr308) antibody, β-tubulin antibody and β-actin antibody were purchased from Bioss (Bioss Beijing, China). KN-62 was purchased from TOCRIS (Minneapolis, MN, USA). Mandarin fish *S. chuatsi* (3 months of age, total length 21.22 ± 1.35 cm) were obtained from the Chinese Perch Research Center of Huazhong Agricultural University (Wuhan, China). Fish were maintained in an aquarium (60 × 45 × 45 cm) with a continuous system of water filtration and aeration at constant temperature (25 ± 1 °C). Fish were acclimated to the condition for 2 weeks prior to experimentation. Fish were fed once daily at 5:30 pm with live India mrigal *Cirrhinus mrigala* juvenile. The animal protocol was approved by The Scientific Ethics Committee of Huazhong Agricultural University (Wuhan, China). The ethical code is HZAUFI-2019-013.

### 4.2. Learning Test

The experimental fish were randomly divided into two groups (*n* = 12)—the control group and the learning group. The control group: Experimental fish were paired with the negative demonstration fish (fish without training). The learning group: Experimental fish were paired with the positive demonstration fish (fish had been pre-trained to accept dead prey fish using the method mentioned in the previous study [15]). The live and frozen India mrigal were used as live prey fish and dead prey fish, respectively. During the test period, mandarin fish were fed with prey fish 10 times (2 min per time) at 5:30 pm each day over a 12 day period. The prey fish were placed 20 cm away from mandarin fish, and only one live or dead prey fish was allowed into each aquarium at any time. After 2 min, the live or dead prey fish that had not been eaten were removed from the aquarium. The testing process was recorded with a digital camera. The success rates, learning times of feeding dead prey fish, and food intake were counted from the videos. As long as one dead prey fish was successfully accepted, the experimental fish was marked as a successful transformer of feeding habits to dead prey fish. No mandarin fish died of natural causes during the testing. 

### 4.3. Sample Collection

The experimental fish (*n* = 12) were anesthetized with MS-222 (Argent Chemical Laboratories, Redmond, WA, USA) (200 mg L^−1^), and euthanized by decapitation at 2 h after feeding. Brain tissues were immediately frozen in liquid nitrogen upon surgical resection and stored at −80 °C prior to RNA isolation and protein isolation. 

### 4.4. RNA Isolation and Reverse Transcription

Total RNA was isolated with a Trizol reagent (TaKaRa, Tokyo, Japan) following the manufacturer’s protocol. The extracted RNA was re-suspended in 30 μL of RNase-free water and then quantified with a BioTek Synergy™2 Multi-detection Microplate Reader (BioTek Instruments, Winooski, VT, USA) and agarose gel electrophoresis. One microgram of total RNA was used for reverse transcription with Revert Aid™ Reverse Transcriptase (TaKaRa, Tokyo, Japan) according to the manufacturer’s instructions. The synthesized cDNA was stored at −20 °C.

### 4.5. Real-Time Quantitative PCR

Primers were designed using Primer 5.0 software (Premier, Canada) based on the sequences obtained from transcriptome sequencing data of mandarin fish, and they were synthesized by Sangon (Shanghai, China). The amplification information of the primers is listed in Table 2. A set of housekeeping genes including β-actin, *rpl13a*, *b2m*, *ywha2*, *hmbs* and *sdha* were selected according to the literature [48]. The *Rpl13a* gene was more stable and amplified as the internal control. Real-time quantitative PCR was carried out with a MyiQ™ 2 Two-Color Real-Time PCR Detection System (Bio-Rad, Hercules, CA USA). PCR was performed in a total volume of 20 μL containing 10 μL of AceQ® qPCR SYBR® Green Master Mix (Vazyme, Piscataway, NJ, USA), 0.4 M of each primer, and 20 ng of cDNA. The PCR cycling parameters were 95 °C for 5 min, followed by 40 cycles of 95 °C for 10 s, and annealing temperatures for 30 s. A melt curve analysis was performed from 65 to 95 °C, gradually increasing at 0.5 °C/6 s, to verify the specificity. Reactions for each sample were performed in triplicate. Gene expression levels were quantified relative to the expression of *rpl13a* using the optimized comparative Ct (2^−ΔΔ*C*t^) value method [49]. Data are presented as mean ± S.E.M. (*n* = 6).

### 4.6. Transcriptome Sequencing

An equal amount of total RNA of each group was used to construct the libraries for the transcriptome analysis using MGIEasy RNA kit following manufacturer’s instructions (BGI, Wuhan, China). Poly(A) mRNA was purified from total RNA using oligo-dT-attached magnetic beads. Paired-end cDNA libraries were sequenced using the BGISEQ-500 system (BGI, Wuhan, China). Image deconvolution and base calling were performed with the SOAPnuke. Clean reads were obtained by removing adaptor reads and low quality reads (Q ≤ 10), on which all following analyses were based. Transcriptome assembly was carried out with short reads assembling program Trinity with k-mer length 25 bp. The reads were mapped back to assembled contigs. By using the paired-end information, it was able to detect contigs from the same transcript as well as the distances between these contigs. We connected the contigs using N to represent unknown sequences between each pair of contigs, and then scaffolds were made. Paired-end reads were used again for gap filling of scaffolds to obtain sequences with least Ns and could not be extended on either end. Such sequences were defined as unigenes. To annotate the transcriptome, we performed the BLAST alignment between unigene and databases such as the Kyoto Encyclopedia of Genes and Genomes (KEGG), Gene Ontology (GO), NR, NT, SwissProt, Pfam and KOG with Blast2GO, hmmscan and getorf software. 

To estimate expression levels, the RNA-Seq reads generated were mapped to the unigenes using Bowtie2. Gene expression levels were measured through RSEM. We analyzed the differentially expressed genes used DEGseq method described before [50], and false discovery rate (FDR) ≤ 0.001 and fold change ≥ 2.00 as the threshold to judge the significance of gene expression difference. A GO functional analysis and a KEGG pathway analysis were then carried out in differentially expressed genes.

### 4.7. Signaling Pathways Analysis

Brain tissues stored at −80 °C were solubilized in a lysis buffer, and lysates were separated on 10% SDS-PAGE gel. Proteins were then transferred onto a PVDF membrane. Phospho-cAMP-response element-binding protein (p-CREB), Phospho-protein kinase A (p-PKA), Phospho- extracellular regulated protein kinases (p-ERK1/2), Phospho-protein kinase B (p-AKT), Phospho-Ca^2+^/calmodulin-dependent protein kinase II (p-CaMKII), Phospho-S6 ribosomal protein (p-S6), β-tubulin and β-actin were detected by immunoblotting with the antibody (1:1000–1:4000). Blots were probed by goat anti-rabbit or goat anti-mouse second antibody with IR-Dye 680 or 800 cw labeled (1:2000–1:4000, Licor, Lincoln, NE, USA) at room temperature for 1 h. The membranes were then visualized using a LiCor Odyssey scanner (Licor, Lincoln, NE, USA) and quantified with ImageJ 1.44 software (National Institute of Health, Bethesda, MD). The phosphorylation levels of the cAMP-response element-binding protein (CREB), the cAMP-dependent protein kinase (PKA), the extracellular regulated protein kinases (ERK1/2), protein kinase B (AKT), Ca^2+^/calmodulin-dependent protein kinase II (CaMKII) and S6 ribosomal protein (S6) were normalized according to the loading of proteins by expressing the data as a ratio over β-actin or β-tubulin.

### 4.8. Effects of the Inhibitors for PKA and CaMKII Signaling Pathway

Brain cells were obtained from mandarin fish and maintained at 28 °C in L15 (leibovitz’s L15-medium) containing 20% newborn calf serum and 5% bio-antibiotics (Penicillin-Streptomycin). After three generations, cells were plated on 60 mm dishes and incubated at 28 °C without CO_2_. Four days later, the cells were with 80% coverage, were washed twice, and starved for 24 h with L15 media. The starved cells were treated for 0.5–3.0 h with either buffer alone or different concentration of the inhibitors (H-89 for the PKA signaling pathway, KN-62 for the CaMKII signaling pathway). The best time point and drug concentration were chosen based on pre-experiment. Total RNA or protein samples were collected with a lysis buffer or Trizol reagent. The gene expression levels of learning or appetite control related genes and the phospho-PKA, phospho-CaMKII and phospho-CREB were detected by real-time quantitative PCR and a Western blotting analysis, respectively.

### 4.9. ChIP Assay

A CHIP assay was performed using a SimpleChIP® Enzymatic ChIP kit (CST, USA) according to the manufacturer’s instruction. The brain tissue from KM mice was isolated, and the DNA sample was purified with a ChIP Kit following manufacturer’s instructions. Positive controls were 10% of each DNA sample, which did not include the immunoprecipitation step. The remainder of the samples were equally divided into two groups. The experimental group was immunoprecipitated with a specific mouse c-Fos antibody (Santa Cruz, USA; 1:500), whereas the negative control group was immunoprecipitated with a general mouse IgG antibody (proteintech, wuhan; 1:250). The isolated DNA was firstly purified then amplified by PCR using specific primers encompassing the AP-1 binding site of the anorexigenic neuropeptides *proopiomelanocortin* (*pomc*) gene. The putative transcription factor binding sites (TFBS) of the regulatory region at 2000 bp upstream from the transcriptional start site of the *pomc* gene were predicted by AliBaba2.1 software (http://gene-regulation.com/pub/programs/alibaba2) and PROMO using version 8.3 of TRANSFAC (http://alggen.lsi.upc.es/cgi-bin/promo_v3/promo/promoinit.cgidirDB=TF_8.3). These putative binding sites of c-Fos protein are located at -1541 to -1534 bp (gtttcagt, site 1), -1128 to -1119 bp (tcactgaagg, site 2), and -383 to -374 bp (tttatagtga, site 3) on the upstream of the transcriptional start site. The primers are 5′-GCCAACCCAAACTTACCT-3′ (site 1, forward primer) and 5′-TTCAACCTCCCATCCACC-3′ (site 1, reverse primer), 5′-AACTGGGAGATGATGGGG-3′ (site 2, forward primer) and 5′-CGGAGTGACTTCCTGCTGT-3′ (site 2, reverse primer), 5′-CCCACTCCAAAAGGTAGC-3′ (site 3, forward primer) and 5′-TGAGGAAGGGGATTTGTTA-3′ (site 3, reverse primer).

### 4.10. Statistical Analysis

Statistical analyses were performed with SPSS 19.0 software (SPSS, Chicago, IL, USA). All data were tested for normality and homogeneity of variances using the Shapiro–Wilk’s test and Levene’s test, respectively. Significant differences were found using a one-way analysis of variance (ANOVA), followed by Fisher’s least significant difference post hoc test and Duncan’s multiple range tests, after confirming data normality and homogeneity of variances. Differences were considered to be significant if *p* < 0.05.

## Figures and Tables

**Figure 1 ijms-20-04399-f001:**
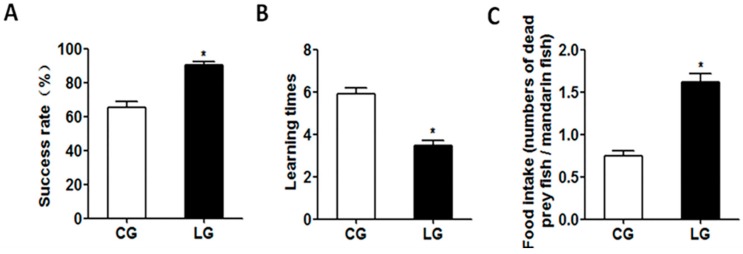
The success rate (**A**), learning times (**B**) and food intake (**C**) of mandarin fish (CG) with the negative demonstration fish (wild mandarin fish without training) and fish (LG) with the positive demonstration fish (pre-trained mandarin fish which could accept dead prey fish). Data are represented as mean ± S.E.M. (*n* = 12). A value followed by * differs significantly from all other values not followed by the same superscript at the same time point based on a one-way analysis of variance (ANOVA) followed by the post hoc test (* *p* < 0.05).

**Figure 2 ijms-20-04399-f002:**
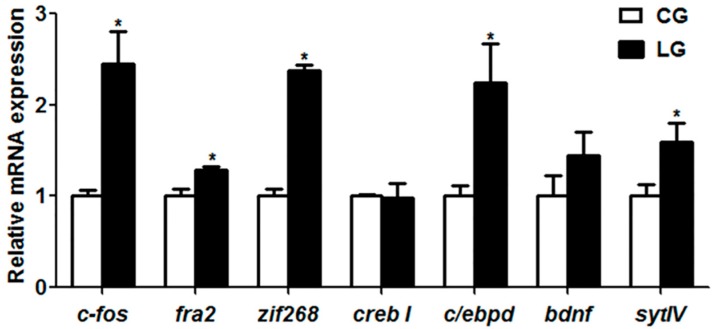
Learning-relative genes expression in mandarin fish (CG) with the negative demonstration fish (wild mandarin fish without training) and fish (LG) with the positive demonstration fish (pre-trained mandarin fish which could accept dead prey fish). Data are represented as mean ± S.E.M. (*n* = 6). A value followed by * differs significantly from all other values not followed by the same superscript at the same time point based on a one-way analysis of variance (ANOVA) followed by the post hoc test (* *p* < 0.05).

**Figure 3 ijms-20-04399-f003:**
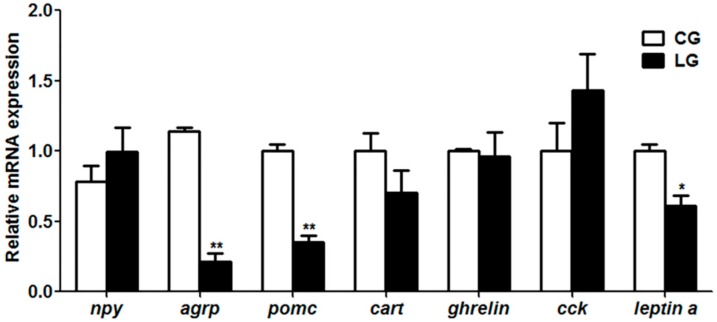
Appetite control genes expression in mandarin fish (CG) with the negative demonstration fish (wild mandarin fish without training) and fish (LG) with the positive demonstration fish (pre-trained mandarin fish which could accept dead prey fish). Data are represented as mean ± S.E.M. (*n* = 6). A value followed by * differs significantly from all other values not followed by the same superscript at the same time point based on a one-way analysis of variance (ANOVA) followed by the post hoc test (* *p* < 0.05, ** *p* < 0.01).

**Figure 4 ijms-20-04399-f004:**
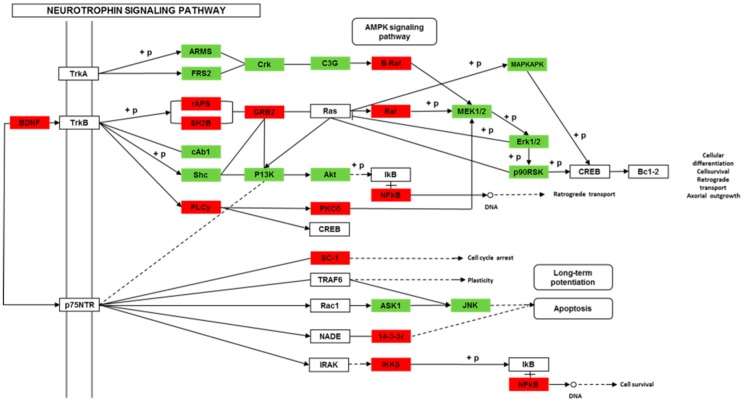
Differentially expressed genes in the neurotrophin signaling pathway between mandarin fish at CG and LG from a transcriptome analysis. The colors of the rectangles were shaded according to the different expression (red: The mRNA expression levels of fish in Group LG were significantly higher than those in Group CG (FDR ≤ 0.001, the absolute value of log2 (Ratio) ≥ 1); green: The mRNA expression levels of fish in Group LG were significantly lower than those in Group CG (FDR ≤ 0.001, the absolute value of log2 (Ratio) ≥ 1)).

**Figure 5 ijms-20-04399-f005:**
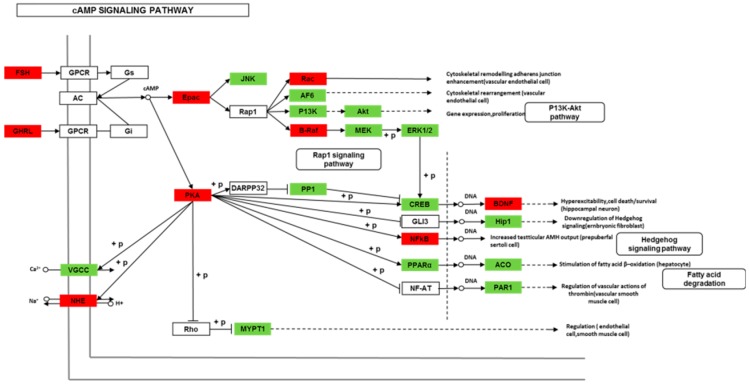
Differentially expressed genes in the cyclic AMP (cAMP) signaling pathway between mandarin fish in the control group (CG) and the learning group (LG) from a transcriptome analysis. The colors of rectangles were shaded according to the different expression (red: The mRNA expression levels of fish in Group LG were significantly higher than those in Group CG (FDR ≤ 0.001, the absolute value of log2 (Ratio) ≥ 1); green: The mRNA expression levels of fish in Group LG were significantly lower than those in Group CG (FDR ≤ 0.001, the absolute value of log2 (Ratio) ≥ 1)).

**Figure 6 ijms-20-04399-f006:**
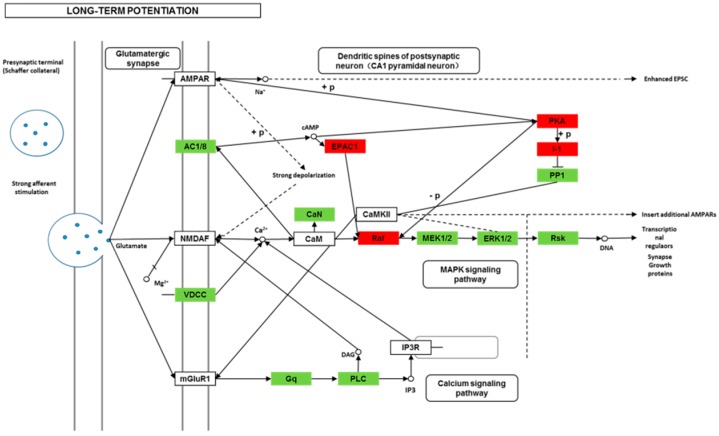
Differentially expressed genes in the long-term potentiation between mandarin fish in the control group (CG) and the learning group (LG) from a transcriptome analysis. The colors of rectangles were shaded according to the different expression (red: The mRNA expression levels of fish in Group LG were significantly higher than those in Group CG (FDR ≤ 0.001, the absolute value of log2 (Ratio) ≥ 1); green: The mRNA expression levels of fish in Group LG were significantly lower than those in Group CG (FDR ≤ 0.001, the absolute value of log2 (Ratio) ≥ 1)).

**Figure 7 ijms-20-04399-f007:**
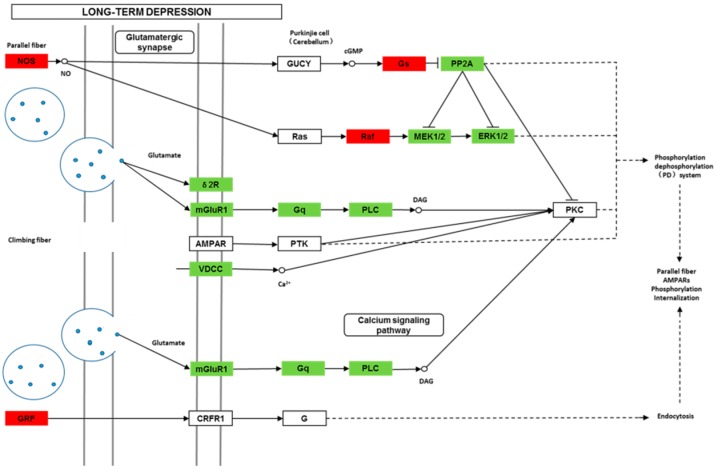
Differentially expressed genes in the long-term depression between mandarin fish in the control group (CG) and the learning group (LG) from a transcriptome analysis. The colors of rectangles were shaded according to the different expression (red: The mRNA expression levels of fish in Group LG were significantly higher than those in Group CG (FDR ≤ 0.001, the absolute value of log2 (Ratio) ≥ 1); green: The mRNA expression levels of fish in Group LG were significantly lower than those in Group CG (FDR ≤ 0.001, the absolute value of log2 (Ratio) ≥ 1)).

**Figure 8 ijms-20-04399-f008:**
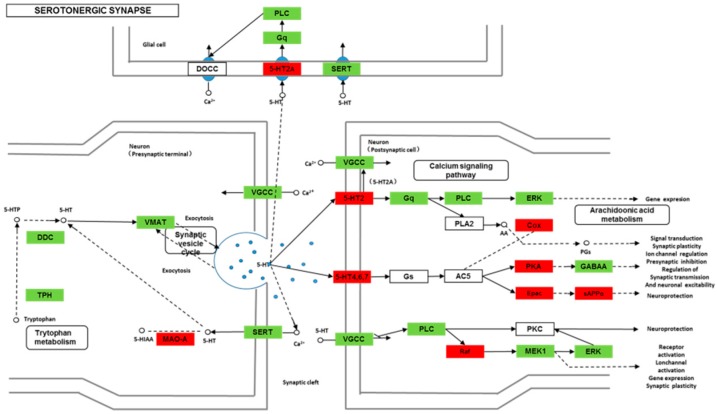
Differentially expressed genes in the serotonergic synapse between mandarin fish in the control group (CG) and the learning group (LG) from a transcriptome analysis. The colors of rectangles were shaded according to the different expression (red: The mRNA expression levels of fish in Group LG were significantly higher than those in Group CG (FDR ≤ 0.001, the absolute value of log2 (Ratio) ≥ 1); green: The mRNA expression levels of fish in Group LG were significantly lower than those in Group CG (FDR ≤ 0.001, the absolute value of log2 (Ratio) ≥ 1)).

**Figure 9 ijms-20-04399-f009:**
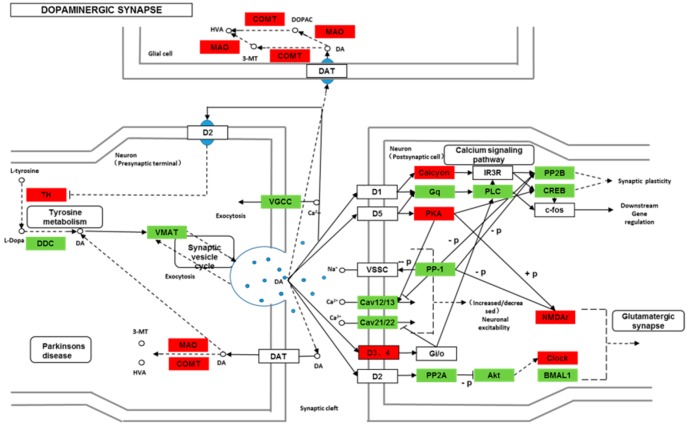
Differentially expressed genes in the dopaminergic synapse between mandarin fish in the control group (CG) and the learning group (LG) from a transcriptome analysis. The colors of rectangles were shaded according to the different expression (red: The mRNA expression levels of fish in Group LG were significantly higher than those in Group CG (FDR ≤ 0.001, the absolute value of log2 (Ratio) ≥ 1); green: The mRNA expression levels of fish in Group LG were significantly lower than those in Group CG (FDR ≤ 0.001, the absolute value of log2 (Ratio) ≥ 1)).

**Figure 10 ijms-20-04399-f010:**
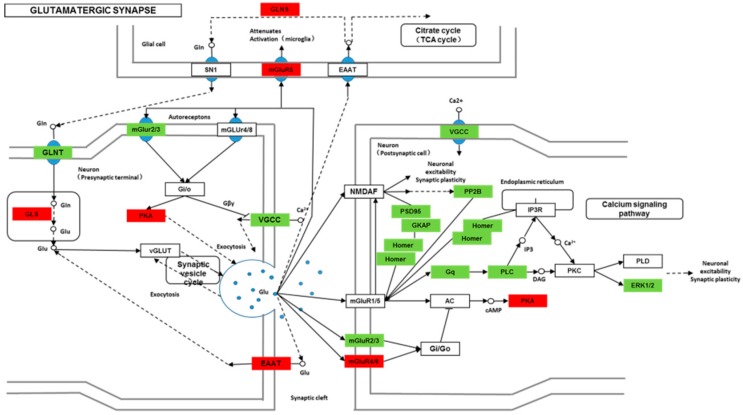
Differentially expressed genes in the glutamatergic synapse between mandarin fish in the control group (CG) and the learning group (LG) from a transcriptome analysis. The colors of rectangles were shaded according to the different expression (red: The mRNA expression levels of fish in Group LG were significantly higher than those in Group CG (FDR ≤ 0.001, the absolute value of log2 (Ratio) ≥ 1); green: The mRNA expression levels of fish in Group LG were significantly lower than those in Group CG (FDR ≤ 0.001, the absolute value of log2 (Ratio) ≥ 1)).

**Figure 11 ijms-20-04399-f011:**
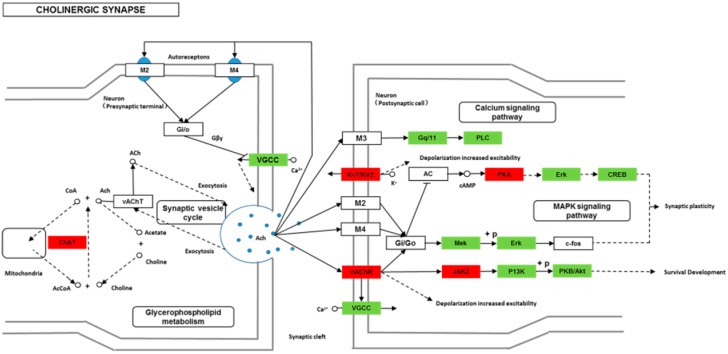
Differentially expressed genes in the cholinergic synapse between mandarin fish in the control group (CG) and the learning group (LG) from a transcriptome analysis. The colors of rectangles were shaded according to the different expression (red: The mRNA expression levels of fish in Group LG were significantly higher than those in Group CG (FDR ≤ 0.001, the absolute value of log2 (Ratio) ≥ 1); green: The mRNA expression levels of fish in Group LG were significantly lower than those in Group CG (FDR ≤ 0.001, the absolute value of log2 (Ratio) ≥ 1)).

**Figure 12 ijms-20-04399-f012:**
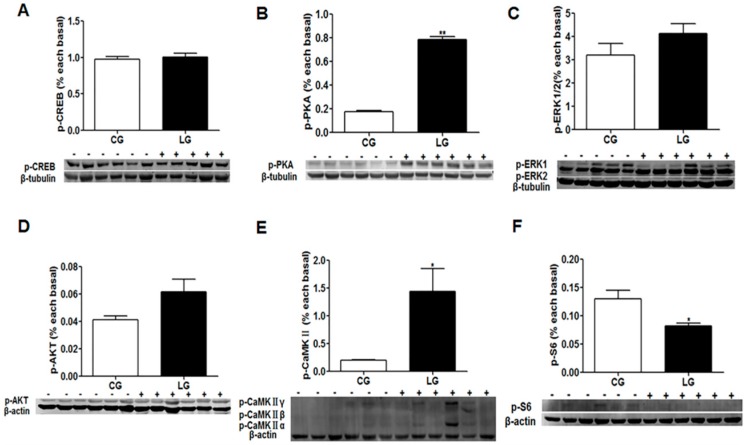
The phosphorylation levels of cAMP-response element binding (CREB) (**A**), protein kinase A (PKA) (**B**), extracellular regulated protein kinases (ERK)1/2 (**C**), AKT (**D**), Ca^2+^/calmodulin-dependent protein kinase II (CaMKII) (**E**) and S6 ribosomal protein (S6) (**F**) of mandarin fish (CG) with the negative demonstration fish (wild mandarin fish without training), and fish (LG) with the positive demonstration fish (well-trained fish which could accept dead prey fish). Data are represented as mean ± S.E.M. (*n* = 6). A value followed by * differs significantly from all other values not followed by the same superscript at the same time point based on a one-way analysis of variance (ANOVA) followed by the post hoc test (* *p* < 0.05, ** *p* < 0.01).

**Figure 13 ijms-20-04399-f013:**
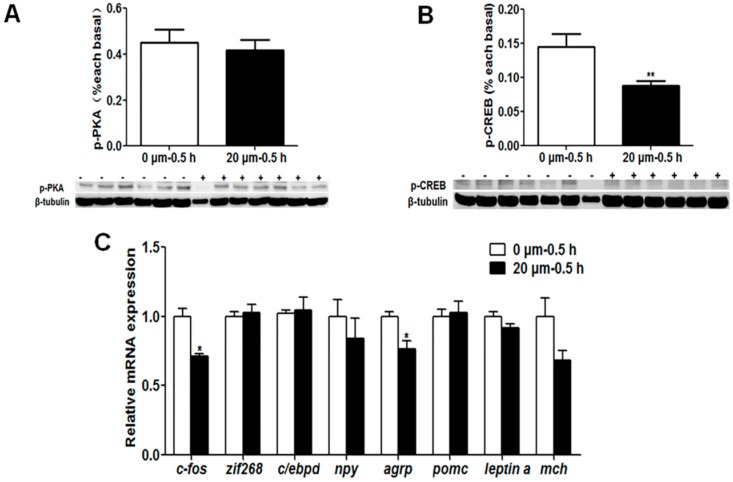
The phospho-protein kinase A (p-PKA) (**A**) and p-CREB levels (**B**), and learning and appetite control-relative genes expression (**C**) in mandarin fish brain cells with the treatment of inhibitor H-89 (20 μm, 0.5 h). Data are represented as mean ± S.E.M. (*n* = 6). A value followed by * differs significantly from all other values not followed by the same superscript at the same time point based on a one-way analysis of variance (ANOVA) followed by the post hoc test (* *p* < 0.05, ** *p* < 0.01).

**Figure 14 ijms-20-04399-f014:**
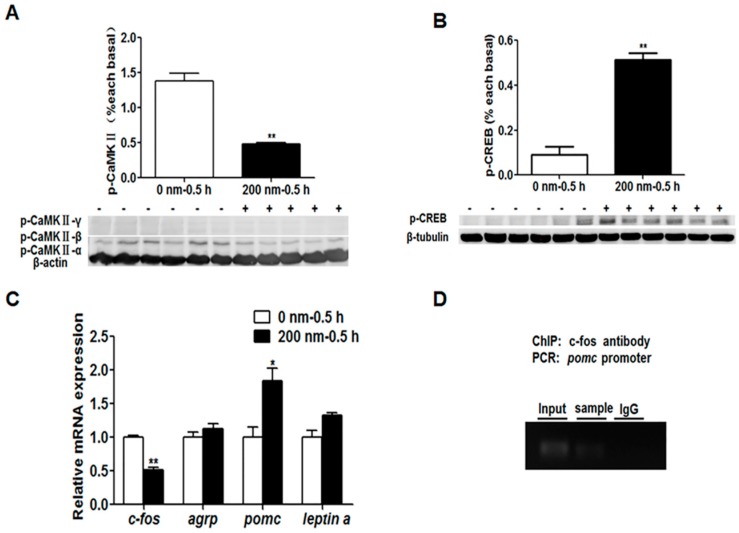
The p-CaMKII (**A**) and p-CREB levels (**B**), and learning and appetite control-relative genes expression (**C**) in mandarin fish brain cells with the treatment of inhibitor KN-62 (200 nm, 0.5 h). Chromatin immunoprecipitation (ChIP) assays with anti-c-fos antibody or control mouse IgG (**D**) for the determination of c-Fos binding sites in the *pomc* promter in the brain tissue of KM mice (Kunming mice). Data are represented as mean ± S.E.M. (*n* = 6). A value followed by * differs significantly from all other values not followed by the same superscript at the same time point based on a one-way analysis of variance (ANOVA) followed by the post hoc test (* *p* < 0.05, ** *p* < 0.01).

**Table 1 ijms-20-04399-t001:** Summary of data generated from mandarin fish transcriptome.

Sample	Total Number	Total Length (bp)	Mean Length (bp)	N50 (bp)	N70 (bp)	N90 (bp)	GC (%)
**C1**	64,501	60,882,979	943	1551	931	383	43.36
**C2**	59,502	44,077,685	740	1076	667	317	42.59
**C3**	60,138	57,556,579	957	1571	953	389	43.33
**L1**	56,441	44,437,838	787	1170	736	334	42.69
**L2**	66,283	53,448,676	806	1209	748	342	42.9
**L3**	65,094	55,146,532	847	1320	816	353	42.91
**All-Unigene**	93,699	98,332,054	1049	1775	1080	430	43.30

**Table 2 ijms-20-04399-t002:** Gene-specific primers used for the analysis of gene expression in mandarin fish.

Gene	Primers	Sequence 5′-3′	Product Size (bp)	Annealing Temp (°C)
*rpl13a*	*rpl13a*-F	TATCCCCCCACCCTATGACA	100 bp	59
	*rpl13a*-R	ACGCCCAAGGAGAGCGAACT		
*c-fos*	*c-fos*-F	CGATGATGTTTACCGCTTTC	88 bp	60
	*c-fos*-R	TAGTATCCCAGATTGTCCC		
*bdnf*	*bdnf*-F	AACTGCCCTCACTCACA	107 bp	58
	*bdnf*-R	ACCTCCCTGGCTCTTAT		
*fra2*	*fra2*-F	CAACCAGGACCTCCAGTG	215 bp	58
	*fra2*-R	TCTACGCCTTTCAATCTC		
*sytIV*	*sytIV*-F	TGTCGGAGGATTAGAACG	133 bp	58
	*sytIV*-R	CTGAAAGTCCAATGGGTAC		
*crebI*	*crebI*-F	ATACACCCTCCCACTTCA	97 bp	58
	*crebI*-R	TCTCCTCCACATCCGTTC		
*zif268*	*zif268*-F	GGATCTTGCCGTGCCTCTTG	221 bp	60
	*zif268*-R	CTGCGACCGCCGTTTCTC		
*c/ebpd*	*c/ebpd*-F	GCAGGAGAAGGCGGATTT	88 bp	60
	*c/ebpd*-R	CTGGGAAGGCAGGGATGA		
*pomc*	*pomc*-F	TGTTAGTGGTGGTGATGGC	268 bp	58
	*pomc*-R	CTGTCGCTGTGGGCTTTC		
*npy*	*npy*-F	GGAAGGATACCCGGTGAAA	202 bp	52
	*npy*-R	TCTTGACTGTGGAATCGTG		
*agrp*	*agrp*-F	GTGCTGCTCTGCTGTTGG	295 bp	65
	*agrp*-R	AGGTGTCACAGGGGTCGC		
*mch*	*mch*-F	AAGAAACTCATCCACGAAG	175 bp	52
	*mch*-R	GGTGAAAGTATCCTGCTCC		
*cart*	*cart*-F	TCTGCACGAAGTGTTGGA	176 bp	56
	*cart*-R	GCACATCTTCCCGATACGA		
*ghrelin*	*ghrelin*-F	TTGCTGGTCTTCCTGTTGTGT	101 bp	54
	*ghrelin*-R	TTCCCCTTGTTCTGAGGTTTT		
*cck*	*cck*-F	GTGTTCTCCTGGCTGCTGTG	180 bp	59
	*cck*-R	CCTGCGGCTGGTTGTAGTTG		
*leptin a*	*leptin a*-F	CCTCTGCCAGTGGAAGTA	188 bp	61
	*leptin a*-R	GTGTCAGAGATCAGGCTGT		

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
