# Peer review of "Social Learning of Acquiring Novel Feeding Habit in Mandarin Fish (*Siniperca chuatsi*)"

_ijms, 2019, doi:10.3390/ijms20184399_

Round 1
Reviewer 1 Report
More information about mandarin fish should be included in the introduction.
Line 108-111: it is not clear that how many fish were used for sampling?
Author Response
More information about mandarin fish should be included in the introduction.Answer: Thank you for pointing this out. We have added the information about mandarin fish as your suggestion (Line 60-61).
Answer: Thank you for pointing this out. We have added the sampling information as your suggestion (Line 112).
Reviewer 2 Report
The article entitled ”Social learning of acquiring novel feeding habit in mandarin fish (Siniperca chuatsi)” is well written and provide new knowledge regarding fish learning behaviour. Especially, the molecular pathway of learning procedure in the fish organism becomes better revealed.
The title is appropriate.
The abstract is informative and well compacted as for such complex study.
The introduction is well written and provides good background for the readers.
Materials and methods are described in details enough to ensure their quality.
Results are clearly presented and have great scientifical importance.
Discussion is well written and “touches” all from the important findings from the manuscript.
The only weak point of the manuscript is references. The formatting of references is not of the style adopted for the IJMS journal. Especially authors should correct:
The journal names should be given in abbreviations with dots.
If journal have no abbreviation (one world journals) do not place dot after the journal name.
Date of publication should be bolded.
I found this problem in all references, so please correct it.
Author Response
The article entitled “Social learning of acquiring novel feeding habit in mandarin fish (Siniperca chuatsi)” is well written and provide new knowledge regarding fish learning behaviour. Especially, the molecular pathway of learning procedure in the fish organism becomes better revealed. The abstract is informative and well compacted as for such complex study.
The introduction is well written and provides good background for the readers.
Materials and methods are described in details enough to ensure their quality.
Results are clearly presented and have great scientifical importance.
Discussion is well written and “touches” all from the important findings from the manuscript.
The only weak point of the manuscript is references. The formatting of references is not of the style adopted for the IJMS journal. Especially authors should correct:
The journal names should be given in abbreviations with dots.
If journal have no abbreviation (one world journals) do not place dot after the journal name. Date of publication should be bolded. I found this problem in all references, so please correct it.
Answer: Thank you for pointing this out. We have reformatted the references as your suggestion.
Reviewer 3 Report
The authors are applauded on the extent of their work with molecular tests and mechanisms associated with social learning habits in fish.
Although social learning plays an important role in gaining new foraging skills and food preferences, your studies show that molecular mechanisms in addition to social learning are associated with new feeding habits in fish. In this regard, fish may be a model for studying feeding and food habits in other animals and humans.
In this regard, I suggest that the authors discuss how their findings with their molecular studies with fish would provide methods to improve the feeding and food habits and preferences of other animals and livestock of economic importance, and for better health preferences in humans.
Author Response
The authors are applauded on the extent of their work with molecular tests and mechanisms associated with social learning habits in fish.
Although social learning plays an important role in gaining new foraging skills and food preferences, your studies show that molecular mechanisms in addition to social learning are associated with new feeding habits in fish. In this regard, fish may be a model for studying feeding and food habits in other animals and humans.
In this regard, I suggest that the authors discuss how their findings with their molecular studies with fish would provide methods to improve the feeding and food habits and preferences of other animals and livestock of economic importance, and for better health preferences in humans.
Answer: Thank you for pointing this out. We have discussed our findings about its importance for improving the feeding, food habits and preferences of other animals and livestock as your suggestion (Line 504-508).